

# Fire Weather Compromises Forestation-reliant Climate Mitigation Pathways

Felix Jäger[1], Jonas Schwaab[1], Yann Quilcaille[1], Michael Windisch[1,2], Jonathan Doelman[3,4],
Stefan Frank[5], Mykola Gusti[5], Petr Havlik[5], Florian Humpenöder[2], Andrey Lessa Derci Augustynczik[5],
Christoph Müller[2], Kanishka B. Narayan[6], Ryan S. Padrón[1,7], Alexander Popp[2], Detlef van Vuuren[3,4],
Michael Wögerer[5], and Sonia I. Seneviratne[1]

[1]Institute for Atmospheric and Climate Science, ETH Zurich, Zurich, Switzerland
[2]Potsdam Institute for Climate Impact Research, Member of the Leibniz Association, Potsdam, Germany
[3]PBL Netherlands Environmental Assessment Agency, The Hague, The Netherlands
[4]Copernicus Institute for Sustainable Development, Utrecht University, Utrecht, The Netherlands
[5]International Institute for Applied Systems Analysis, Laxenburg, Austria
[6]Joint Global Change Research Institute, Pacific Northwest National Laboratory, College Park, USA
[7]Swiss Federal Institute for Forest, Snow and Landscape Research WSL, Birmensdorf, Switzerland

**Correspondence:** Felix Jäger (felix.jaeger@env.ethz.ch)

**Abstract.** Forestation can contribute to climate change mitigation. However, an increasing frequency and intensity of climate extremes is posed to have profound impact on forests, and consequently on the mitigation potential of forestation efforts. In this perspective, we critically assess forestation-reliant climate mitigation scenarios from five different Integrated Assessment Models (IAMs) by show-casing the spatially explicit exposure of forests to fire weather and the simulated increase in global annual burned area. We provide a detailed description of the feedback from climate change to forest carbon uptake in IAMs. Few IAMs are currently accounting for feedback mechanisms like loss from fire disturbance. Consequently many forestation areas proposed by IAM scenarios will be exposed to fire-promoting weather conditions and without costly prevention measures might be object to frequent burning. We conclude that the actual climate mitigation portfolio in IAM scenarios is subject to substantial uncertainty and that the risk of overly optimistic estimates of negative emission potential of forestation should be avoided. As a way forward we propose how to integrate more detailed climate information when modeling climate mitigation pathways heavily relying on forestation.

## 1 Introduction

Negative emissions, i.e. carbon uptake from the atmosphere, are essential for ambitious climate change mitigation (van Vuuren et al., 2017). Integrated Assessment Models (IAMs) are commonly used to derive emission scenarios compatible with maximum warming levels of 1.5 or well-below 2.0 °C of global warming relative to pre-industrial levels (IPCC, 2018), which were set as targets as part of the Paris Agreement (UNFCCC, 2015). These simulations, which provided forcing datasets for greenhouse gas (GHG) emissions and land use in the Coupled Model Intercomparison Project 6 (CMIP6, see Fig. 2), include a substantial amount of carbon removal from the atmosphere for reaching net zero GHG emissions within the next 20 to 40





years. In addition to bioenergy with carbon capture and storage (BECCS, Vaughan et al., 2018), the assessed IAMs project
afforestation and reforestation (forestation, A/R) over an area ranging from 2.6 to 14 million km$^2$ (Fig. 1, suppl. Fig. S9, Popp
et al., 2017) as a nature-based carbon storage strategy. This considerable spread across an order of magnitude is also present in
terms of carbon sequestration potential (approximately 0.5 to 12 GtCO$_2$a$^{-1}$ by 2100) (Griscom et al., 2017; Fuss et al., 2018).
Sources of uncertainty lie both in future socio-economic dynamics (Popp et al., 2017; Brown et al., 2021) and vegetation
responses to disturbances inducing those that are induced or exacerbated by climate extremes (Anderegg et al., 2020).

Fire as one prominent and large hazard for carbon accumulation in forests is influenced by the increasing intensity and
frequency of climate extremes (Seneviratne et al., 2021). Generally, wildfire with increasing frequency and intensity is reducing
biomass and soil carbon stocks (Walker et al., 2019; Pellegrini et al., 2022), limiting long-term carbon uptake (Koch and
Kaplan, 2022). Furthermore, fire in some places is a key factor determining whether forests can exist or not (Murphy and
Bowman, 2012) and can substantially reduce tree and forest cover (Lasslop et al., 2020). Intensifying fire weather as observed
(Abatzoglou et al., 2018; Jones et al., 2022) and projected (Son et al., 2021) hence puts increasing pressure on forests and
their associated carbon storage. Thus, future fire regimes might significantly influence whether long-term global-scale carbon
capturing by A/R is feasible in the projected ranges, with the projected costs and speeds. Comparing maps of fire weather
change and forestation potential Hermoso et al. recently brought forward their concern on increasing fire disturbance of forest
restoration projects in Europe. Clarke et al. recently illustrated how high water vapor pressure deficit might foster forest fires,
and thus might endanger carbon sinks. Golub et al. (2022) have put forward a land use allocation assessment using die-back
rates per biome changing along global mean temperature showing that fire might not compromise forest based climate solutions.
While it is well-known that fire can substantially reduce the mitigation potential of forests (Anderegg et al., 2020; Lasslop et al.,
2020; Wang et al., 2015), it is unclear how IAMs account for the effects of increasing climate extremes and likelihood of fires. A
typical channel for such natural hazards to be included in IAMs is via adjusted long-term carbon stock potential of a certain land
use types. Since IAMs are broadly used for emission projections informing climate model experiments as well as international
climate policies, it is essential to better understand and quantify uncertainties in their modeling process and assumptions about
the potential for A/R.

    In this perspective, we therefore assess whether the effectiveness of forestation in the current spatially explicit representation
of IAM projections could be compromised by fire disturbance, based on available scenarios from a range of state-of-the-art
IAMs. We ask: (1) What is the amount of forestation as projected by IAMs and how sensitive is it to climate change effects? (2a)
How much and which areas increase the exposure of global forests to fire weather and (2b) how do exposure change, i.e. forest
expansion, and hazard change, i.e. fire intensification from climate change, compare as drivers of forest fire danger rise? (3)
How is spatially explicit information about forest disturbances and climate change treated in state-of-the-art land use allocation
in IAMs? (4) How can the representation of climate impacts on forestation be improved to arrive at more substantiated climate
mitigation scenarios.



## 2 Large-scale forestation in ambitious climate mitigation scenarios

IAMs project global forest area to be expanded vastly under SSP1-1.9 and SSP1-2.6 (Fig. 1), which are roughly compatible with maximum warming levels of 1.5 and 2 °C of global mean surface temperature above pre-industrial levels, respectively (building on Shared Socio-economic Pathways (SSP)1, van Vuuren et al., 2017; IPCC, 2018). To arrive at this result, we

assembled several land-use projections to analyze A/R scenarios including the Land Use Harmonization (LUH) data set 2 and five IAM data sets under SSP1-2.6. A detailed description of the setup used in these models (Table C1) and a discussion of inter-model differences in forest cover are given in the Method section.

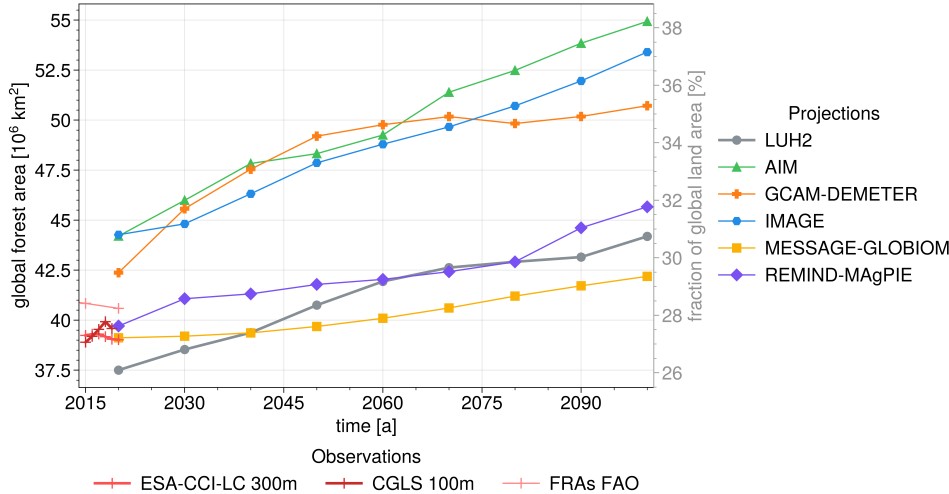

**Figure 1.** Projected and observed (red) global forest area. Six projections under SSP1-2.6 are shown for LUH2 and five different IAMs. Two observational data sets from satellite imagery (Harper et al., 2023; Buchhorn, 2020) and data from forest resource assessments (FRAs) of the FAO (FAO, 2016; FAO, 2020) are shown for past and present.

The global forest area in 2020 in the scenarios is comparable to observational data sets and forest resource assessments (with deviations at the local scale, see also Chen et al., 2020). The forest area in SSP1-2.6 grows from 39 to 44 Mkm$^2$ in 2020 42

to 55 Mkm$^2$ in 2090. This is an expansion of 3 to 10 Mkm$^2$ corresponding to 7 to 22 %. Similar but even stronger trends are found in the SSP1-1.9 projections, especially for IMAGE and AIM (see supplementary Fig. S9).

The forestation pathways considered here are simulated by state-of-the-art IAMs, which provide the scenarios underlying the CMIP6 simulations within the model chain towards Earth System Models (ESMs) (emission and land use change pathways, see Fig. 2). In the typical setup, input data on potential vegetation is used in the land use models within socio-economic models

to simulate land use change within a certain climate mitigation scenario. This land use change is harmonized with data sets on historical land use and land cover change by LUH for the generation of one forcing dataset fed to the land models in ESMs. Along this chain, IAM outputs also influence the main future climate simulations assessed in the 6th Assessment Report of



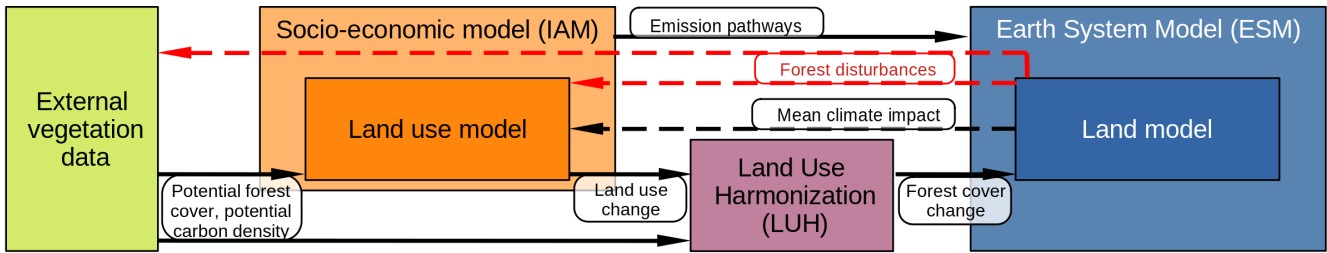

**Figure 2.** Information flow of forest-related data in the typical model chain from IAMs to ESMs for scenarios used in CMIP6. Black arrows indicate established, red arrows the here proposed data exchange in the context of forest fire risk. Dashed arrows indicate partly established links in some of the IAMs. While some IAMs include basic information about the overall fire regime encoded in input data about vegetation potential or impacts by mean climate, most are lacking representation of geographically explicit disturbances regimes changing over time.

the IPCC (IPCC, 2021). In most of the assessed frameworks however, forest-related feedback from climate to land use and vegetation potential modeling is lacking or only implemented partially (red and dashed arrows in Fig. 2).

Accounting for future climate changes – particularly climate extremes – is likely to significantly alter the carbon sequestration potential and the employed forestation in scenarios produced by these IAMs. In a comparison of land use projections with and without all implemented climate impacts, the global forest expansion of MESSAGE-GLOBIOM and REMIND-MAgPIE showed very large sensitivity of 25 % and up to 50 %, respectively (see suppl. Fig. S7). Consequently, both climate-impacted grid-cell level and globally aggregated carbon sequestration potential estimates matter for the overall mitigation portfolio in
ambitious climate change mitigation scenarios.

## 3 Forest fire danger under intensifying fire weather

Here, we use annual values of maximum seasonally averaged (SA) Canadian Fire Weather Index (FWI) (Abatzoglou et al., 2019; Quilcaille et al., 2022), $\text{FWI}_{SA}$, to analyze the fire hazard for forests in IAM projections. FWI combines atmospheric conditions (heat, air dryness, lacking precipitation and wind speed, Wagner, 1987; Wang et al., 2015), which are a crucial driver
of forest fire activity. FWI is one of the most widely used indicators for fire weather, which shows strong links to actual fire impacts (Bedia et al., 2015; Abatzoglou et al., 2018; Jones et al., 2022, see also suppl. section S1). In this study we mainly use FWI because fire weather simulations by ESMs relying only on atmospheric conditions are significantly robuster than ESM fire impact simulations due to limited performance of fire modules within CMIP6 models and dependence on land cover (Hantson et al., 2020; Gallo et al., 2023; Spafford and MacDougall, 2021).
To represent global forest fire danger, we compute the weighted average of $\text{FWI}_{SA}$ over the global forest area, i.e. $\overline{\text{FWI}_{\text{F}}}$ (see appendix A for formula). This global indicator proves to be highly correlated to global burned forest area share. Also locally, FWI is spatially consistently in positive interannual correlation (weighted mean across forest areas R ≈ 0.3) with burned area (Abatzoglou et al., 2018; Bedia et al., 2015; Grillakis et al., 2022; Jones et al., 2022, see suppl. Fig S1 and section S1).



Compared to early industrial times, $\overline{\mathrm{FWI}}_{F,obs}$, computed over constant tree cover of 2020 from the satellite product ESA-CCI-LC (Harper et al., 2023), has substantially risen (Fig. 3, historical). For the period 2020 until 2050 under SSP1-2.6

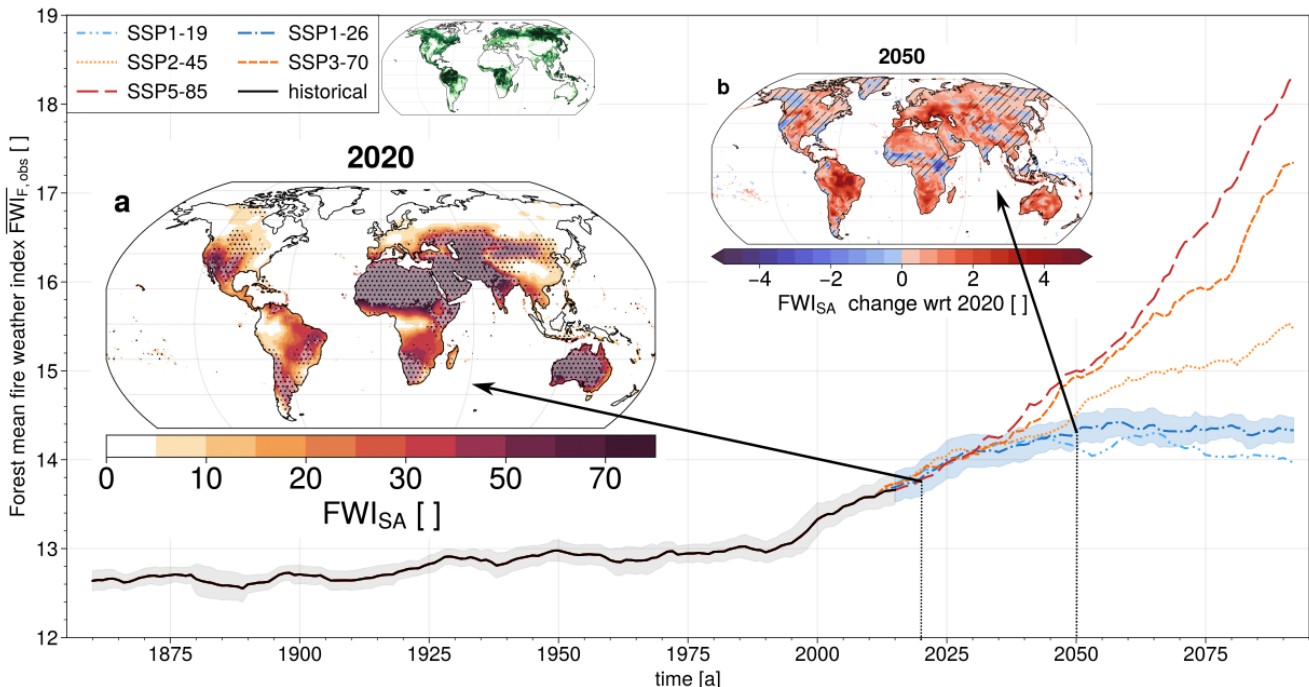

**Figure 3.** Fire weather intensifying over time in the historical simulation and in five future scenarios. The focus of this study is SSP1-2.6 (dark blue, roughly 2°C global warming scenario). The global forest area weighted mean FWI, $\overline{\mathrm{FWI}}_{F,obs}$ is computed over forest area from satellite-observed tree cover (ESA-CCI land cover, Harper et al. (2023)) in 2020 (inset next to legend). Around the lines representing multi-model median the shading indicates typical year-to-year variability (multi-model mean running 10-year standard deviation) of the global aggregate value in the historical and the SSP1-2.6 scenario. Maps show the distribution of the 10-year mean, multi-model median $\mathrm{FWI}_{SA}$ over land in 2020 (a) and its changes under SSP1-2.6 until 2050 (b). Stippling in a) rules out regions of low ($< 5\,\%$) forest cover. Hatching in b) indicates areas where less than eight ESMs out of ten agree on the sign of change, while the global rise of $\overline{\mathrm{FWI}}_{F,obs}$ is found robustly across ESMs.


the available CMIP6 models project a robust increase in fire weather intensity in most land areas, including all of South America, parts of North America, northern and southern Africa, western Eurasia and Australia (Fig. 3b). Under higher warming scenarios, the danger increase would be substantially stronger (Fig. 3, see also Abatzoglou et al., 2019). From 2050 until the end of the century, the five climate scenarios from SSP1-1.9 (slight decline of global danger indicator $\overline{\mathrm{FWI}}_{F,obs}$) to SSP5-8.5

($\overline{\mathrm{FWI}}_{F,obs}$ increase doubles again) diverge substantially, mainly driven by heating.



## 4 Forestation drives forest exposure to extreme fire hazard

Over the course of the century, the projected forested regions in IAMs are increasingly under danger from fire weather (relative increase in $\overline{\text{FWI}_F}$ of up to more than 30 %, Fig. 4a and 4b).

This finding, which is robust against inter-ESM differences in fire weather, has two reasons: First, fire weather intensifies
through heating and drying trends over forest areas that already exist in 2020 as a consequence of climate change (hazard as driver). Second, and more important, most of the increase in danger under SSP1-2.6 is driven by A/R in regions of already high and/or intensifying fire weather (exposure as driver, see Fig. 4c, Fig. S5, S10 and section S2).

This can be attributed to very strong land use policies in SSP1 (Popp et al., 2017), i.e. carbon pricing strongly rewarding expected carbon sequestration on A/R. We found $\overline{\text{FWI}_F}$ to increase over the course of the century in both scenarios SSP1-1.9
and SSP1-2.6 for all data sets. Moreover, $\overline{\text{FWI}_F}$ in projected future afforestation areas is more then 1.3 times the $\overline{\text{FWI}_F}$ for the 2020-forested area (Fig. 4a). Thus, the exposure to intense fire weather in the regions where A/R is projected to occur, drives the overall increase in $\overline{\text{FWI}_F}$.

This increase in danger mainly driven by exposure increases is likely to propagate to fire impacts. Burned forest area is posed to rise under this pressure (according to our analysis by 10-80 %, depending on IAM and ESM projection) (Fig. 4d, e,
see appendix B and supplement for details). Typically, about half of this change can be attributed to A/R in areas of large or increasing annual burned area.

We illustrate A/R regions particularly endangered in the models with the absolute change of the product of hazard and exposure, FWI times $a_F$ over time (Fig. 5). The rise of danger is driven by a regionally varying intersection of up-sloping $\text{FWI}_{SA}$ (hazard-driven) and forest cover share (exposure-driven, see Fig. S5 for spatially explicit signal decomposition and
Fig. S8 for the full signal in 2050).

If areas for forestation were excluded when located in regions of extreme fire regimes (FWI $> 47$ in 2090, 95th percentile of historical conditions), the overall forestation would be reduced by 4 % (IMAGE) to 20 % (AIM) in 2090, clearly illustrating the potential impact of fire danger on forestation allocation.

Generally, a higher FWI regime favors less dense forests and subsequently less effective A/R for negative emissions (Koch
and Kaplan, 2022; Lasslop et al., 2020). Although forests exposed to high FWI are not guaranteed to burn more intensely or more often in every location, long-term and large-scale average response in this direction is very likely, particularly under the absence of fire prevention measures (IAMs do not include the costs for these to date). Such fire prevention may be effective on the medium-term range (Wu et al., 2021), but often unsustainable on longer time scales (Moya et al., 2019; Bowman et al., 2009).

Fire weather and burned area used here are examples to highlight this deficiency. Beyond fire, we expect impacts from other extreme events (e.g. droughts, heatwaves and heavy precipitation) to also play a role in land use decisions (Dale et al., 2001; Johnstone et al., 2016; Seidl et al., 2017; Thom and Seidl, 2016; Hammond et al., 2022). Therefore, we highlight the need to implement more climate impacts, into IAMs, which helps to avoid an overestimation of the negative emission potential of A/R.



**Figure 4.** a) $\overline{\text{FWI}_F}$ for all forests in 2020 (wide boxes) and for A/R areas only in 2050 and 2090 under SSP1-2.6 (upper small boxes). The distribution is given by fire weather projections from 22 Earth System Models. For each IAM, the marker points to the multi-ESM mean, whereas the lines indicate the minimum, 25th, 50th, 75th percentile and the maximum. b) $\overline{\text{FWI}_F}$ and d) $A_{\text{pot.burned}}$ change relative to 2020 under SSP1-2.6 for different forest projections. The shading indicates one standard deviation from the ESM uncertainty of FWI/burned area change. c) and e) Contributions to the relative change in $\overline{\text{FWI}_F}$ and $A_{\text{pot.burned}}$ until 2050 and 2090, from forestation (top), and from fire (weather) change, (bottom) are shown for the six data sets. For readability and interpretability, the range of relative change is cropped to 90 % in b-e. A version with full range including information on the share of annual burned forest area in 2020 is given in Fig. S3. For the corresponding information on $\overline{\text{FWI}_F}$ in SSP1-1.9, see Fig. S10.





**Figure 5.** Absolute change in danger ($FWI_F = FWI \cdot a_F$), the product of fire weather index (hazard) and forest grid cell share (exposure) under SSP1-2.6 in 2090 with respect to 2020 in forestation areas ($\Delta a_F > 5$ %) for LUH2 and five different IAMs. $A_{A/R}$ is the overall forestation area since 2020 and $\Delta_{rel}\overline{FWI_F}$ is the relative change in global forest weighted mean FWI of each model.



## 5 Climate impacts on forests in IAMs to date

None of the modeling frameworks to date explicitly include a broad range of impacts from climate extremes on vegetation. This is troublesome given that increases in climate extremes are expected in all regions, already at 1.5 °C or 2 °C of global warming (Seneviratne et al., 2021). Here, we provide a short overview of whether the analyzed scenarios are produced considering climate effects (feedbacks) for A/R decisions in the associated IAM model versions (see Table 1 and for more details, please refer to appendix C).

**Table 1.** Climate feedbacks into land use decisions in five IAMs for their most recent versions. The versions used in CMIP6 did not include any climate feedbacks (except IMAGE).

| | AIM | GCAM-DEMETER | IMAGE | MESSAGE-GLOBIOM | REMIND-MAgPIE |
|---|---|---|---|---|---|
| Atmospheric $CO_2$ rise on forest allocation | No | Indirectly[*] | No | Indirectly[*] | Yes[**] |
| Atmospheric $CO_2$ rise on forest carbon density | No[†] | No | Yes | No | Offline[**] |
| Change of mean climate on forest allocation | No | Indirectly[*] | No | Indirectly[*] | Yes[**] |
| Change of mean climate on forest carbon density | No[†] | No | Yes | No | Offline[**] |
| Fire disturbances on forest allocation | No | No | No | No | Yes[**] |
| Fire disturbances on forest carbon density | No | No | No[‡] | No | Offline[**] |
| Extreme climate conditions on forests | No | No | Partly[§] | No | Partly[§,**] |

([*] Through impacts on productivity and water availability in agriculture. [**] Through impacts on the potential carbon density, which are provided by an offline simulation using the land model LPJmL. [†] AIM in principle includes this feedback. For the simulations assessed here it was not used for consistency within the multi-IAM mitigation scenario assessment. [‡] In existing forests yes, in forest plantations used for afforestation no. [§] Through extreme temperature and water conditions limiting productivity and enhancing background mortality. Boreal trees can die from heat stress.)

A main entry point for climate feedbacks into IAMs is to modify carbon densities, because the total amount of carbon stored in forests in IAM projections is typically estimated using forest carbon density ($[\rho_C]$ = t/ha) and forest fractional coverage ($[a_F]$ = ha/ha) (Humpenöder et al., 2014, see external vegetation data in Fig. 2). Typically, the carbon content of forests is modeled to monotonously grow towards the determined maximum estimate given by vegetation models, which might include upward or downward impacts from productivity and disturbance shifts under climate change. For long-term and large-scale

averaged results, this provides an approximation relevant for impacts on A/R allocation for carbon uptake. The long-term increase in the carbon stored through A/R is counted as negative emission in IAMs. As a low-cost negative emission option in ambitious mitigation scenarios, A/R allows for residual hard-to-avoid GHG emissions compatible with the target of net-zero emissions. The more areas are available for low-cost A/R potential, the larger is the potential for residual GHG emissions compatible with a low warming target (Humpenöder et al., 2014; Doelman et al., 2020; Frank et al., 2021).

Among the here analyzed models, only IMAGE (existing forests, not in A/R areas) and REMIND-MAgPIE (all forests) account for carbon losses from changing climate conditions including fire in their estimates of forest carbon density. IMAGE and REMIND-MAgPIE include information on climate impacts from the vegetation model LPJmL (Doelman et al., 2020;





Humpenöder et al., 2022), which accounts for day-to-day variability but does not include natural disturbances like fire on this timescale (Schaphoff et al., 2018; Braakhekke et al., 2019). In other IAMs, such processes are not included. Therefore, these

models are expected to produce overly optimistic estimates of A/R effectiveness. Except IMAGE 3.0, the IAM providing the SSP1 simulations (Stehfest et al., 2014; O'Neill et al., 2016), neither the model versions used for projections under CMIP6 nor LUH2 included any climate feedbacks.

The forestation allocation in REMIND-MAgPIE among the other models shows comparably high consistency and flexibility, which is also reflected by its performance to distribute large scale A/R to comparably mild fire conditions (compare $A_{A/R}$ and

$\Delta_{rel}\overline{FWI_F}$ in Fig. 5 and $\Omega$ in Table S1 and section S4). The model shows a comparably small increase in $\overline{FWI_F}$ , partially owing to incorporated climate information, but also due to the incorporation of national tree planting pledges and a high baseline $\overline{FWI_F}(2020)$. Only MESSAGE-GLOBIOM has even smaller $\Delta_{rel}\overline{FWI_F}$, likely stemming from the much smaller A/R volume in that projection.

## 6   Ways forward for climate-related forest disturbances in IAMs

The majority of models used in the scenario framework under CMIP6 are at the beginning of including more climate change impact information into their modeling schemes. In addition to IMAGE and REMIND-MAgPIE, GCAM-DEMETER (Chen et al., 2022; Chen et al., 2019) and MESSAGE-GLOBIOM (Frank et al., 2021) already include some scheme for climate impacts on water availability and productivity in agriculture, but not in forestry (see Table 1 and suppl. Fig. S7, S12 and suppl. section S8). In the MESSAGE-GLOBIOM framework, there are efforts underway to make the forestry model, G4M,

climate informed. Those models that already include detailed climate information, typically find increases in carbon density due to carbon fertilization from increased $CO_2$-levels in the atmosphere. Even though detailed, these estimates might be overly optimistic because they exclude other disturbances leading to carbon losses from explicit modeling, which enables to account for regimes shifts, disturbance interaction and changes over time.

So how can information on disturbances of the carbon pools in forests be propagated into globally modeled land use and

land management? Only one of the assessed models covers the impact of changing carbon sequestration potential on land use decisions. REMIND-MAgPIE assumes a 30-year horizon of foresight of expected future carbon stocks for the distribution of land use including afforestation. Following this example, we plead for a consistent and physically meaningful integration of climate impacts through the effect on carbon density of vegetation. This will also support the tracking of contributions to uncertainty along the model cycle from vegetation to land use to climate back to vegetation (Fig. 2, see supplement for an

example on climate model uncertainty implementation). Sensitivity studies using explicit climate impact representations in IAMs, e.g. modeling carbon density evolution interactively in REMIND-MAgPIE, including probabilistic estimates of climate impacts on land use, would help quantitatively evaluate the sensitivity and reliability of IAM results across spatial and temporal scales. Additionally internalizing the expected increase in costs in fire prevention measures and changes to ecosystem services from e.g. biodiversity could help to make forestation projections more comprehensive (Pawson et al., 2013; Hansen et al.,

2001; Prestele et al., 2016).



Our work highlights the need for improvements of the difficult task of estimating forestation potential in multi-sectoral assessments (on non-technical challenges see e.g. Hollnaicher, 2022) and proposes ways to adress this issue. We showed the forestation potential under SSP1-2.6 modeled in the presented IAM simulations to be severely compromised by fire risk. Including such risk into the assessment will likely diminish the role of forestation in the mitigation portfolio. Given the closing

window of opportunity for limiting global warming to 1.5 °C, these results demonstrate that a climate mitigation strategy minimizing the risks of temperature overshoot must be centered on rapid reduction of carbon emissions.

*Code and data availability.* Land cover satellite products from ESA-CCI and CGLS are available at http://maps.elie.ucl.ac.be/CCI/viewer/download.php and https://land.copernicus.eu/global/products/lc, respectively. Land cover data from GCAM-DEMETER is published along Chen et al. (2020) under https://doi.org/10.25584/data.2020-07.1357/1644253. AIM land use data is available under https://doi.org/10.18959/

20180403.001. Harmonized land use data sets from LUH2 are accessible under https://luh.umd.edu/data.shtml. Land use data from IMAGE, MESSAGEix-GLOBIOM and REMIND-MAgPIE will be made available by the authors. Fire weather index data from CMIP6 along (Quilcaille et al., 2022) is available at http://hdl.handle.net/20.500.11850/583391. Data from CMIP6 are available at https://esgf-node.llnl.gov/search/cmip6/. The used experiments and variables can be found with a search query using: Experiment ID (historical, ssp119, ssp126) and variable (treeFrac, burntFractionAll).

The computer code used for the analysis of forest mean fire weather index as well as a corresponding manual how to apply this code will be made available by the authors on an open access repository.

## Appendix A: Forest area weighted mean fire weather index ($\overline{\text{FWI}_\text{F}}$)

To get a global indicator for forest exposure to fire weather, we calculate a mean FWI weighted by forest fractional area. For fire weather we use an annual indicator of fire season intensity, namely the yearly maximum of the 90 days running average daily

Canadian FWI (Quilcaille et al., 2022). While this index of the seasonal fire weather intensity fits our long-term and global-scale best, our analysis is not sensitive to the use of other measures for fire season intensity or length aggregated from daily FWI data (see e.g. Quilcaille et al. (2022) for such indicators). We are not interested in single extreme days but in the smooth trends of extreme fire hazard from changing atmospheric conditions on a heated planet. This is why the time series of annual values is smoothed with a running 10-year mean such that the 10-yearly maps of forest exposure from IAMs can be matched with

climatic conditions representing changes on the same time scale. Additionally, FWI data were regridded to the 0.5°x0.5° mesh the IAM forest cover data had been assembled on. The FWI is computed from temperature, precipitation, relative humidity and surface wind projections of Earth System Models (ESMs) participating in CMIP6 and providing the necessary variables. For our analysis we either used ten models that had produced simulations not only for SSP1-2.6, SSP2-4.5, SSP3-7.0 and SSP5-8.5, but also for SSP1-1.9, a Tier2 numerical experiment in CMIP6 (Fig. 3), or we used all 22 models providing the SSP1-2.6

simulation (Fig. 4,5). The ten models providing SSP1-1.9 are CanESM5 (50), EC-Earth3 (1), FGOALS-g3 (1), GFDL-ESM4 (1), IPSL-CM6A-LR (6), MIROC-ES2L (10), MIROC6 (3), MPI-ESM1-2-LR (30), MRI-ESM2-0 (5) and UKESM1-0-LL (5), with the respective number of ensemble members in brackets. For the main analysis of danger evolution under SSP1-2.6,



we used ACCESS-CM2 (5), ACCESS-ESM1-5 (40), CMCC-CM2-SR5 (1), CMCC-ESM2 (1), CanESM5 (50), EC-Earth3 (6), FGOALS-g3 (3), GFDL-ESM4 (1), HadGEM3-GC31-LL (1), HadGEM3-GC31-MM (1), INM-CM4-8 (1), INM-CM5-0

(1), IPSL-CM6A-LR (6), KACE-1-0-G (3), MIROC-ES2L (10), MIROC6 (50), MPI-ESM1-2-HR (2), MPI-ESM1-2-LR (30), MRI-ESM2-0 (5), NorESM2-MM (1), TaiESM1 (1), and UKESM1-0-LL (13) to avoid a bias to the model selection. The model mean and median were computed over ensemble mean values per model. We did not account for model similarities with further weighting among ESMs. However, the qualitative findings about the relative change in danger from forest expansion and fire weather intensification of the order of 10 % are not sensitive to model ensemble design choices.

The mean FWI weighted by forest fractional area $a_F$ is then computed by global integration of the product of $FWI_{SA} \in [0, \infty[$ and $a_F \in [0, 1]$

$$\overline{FWI_F}(t) = \frac{\iint_{global} a_F(x, y, t) \cdot FWI_{SA}(x, y, t) \; dA}{\iint_{global} a_F(x, y, t) \; dA} \; , \tag{A1}$$

where dA is the areal increment, a product of the length increments in zonal (x) and meridional (y) direction.

The value of global weighted mean FWI, $\overline{FWI_F}$ was tested to be good predictor of global burned area share. Pearson

correlation is above 0.75 for 5 out of 6 IAMs. The value for MESSAGE-GLOBIOM showing a correlation of still 0.56 could arise from generally low levels of burned area. This suggests that long-term and large-scale signals of global forest area weighted mean FWI change propagate to burned area change (Fig. 4d,e). While this holds for global values, local vulnerability, which is not included in this work, however is expected to significantly modulate the combination of hazard and exposure we present here. For example Zheng et al. (2023) find different response of North American and Siberian boreal forests to climatic

water deficit and extreme temperature suggesting different vulnerabilities.

**Appendix B:  Potentially burned forest area**

To get an estimate of the order of magnitude of change in burned area (BA) from A/R and climate change under SSP1-26, we use the output of land models within CMIP6 ESMs, *BurntFractionAll*, the monthly grid cell share burned in fire. Assuming that an area burned already in one year cannot burn again in the same one, we treat these monthly values as additive to reach annual

values for six different ESMs, namely CESM2 (3), CESM2-WACCM (1), CMCC-CM2-SR5 (1), CMCC-ESM2 (1), CNRM-ESM2-1 (5), and EC-Earth3-Veg (2), with the respective number of ensemble members in brackets. While this approach can give indication of order of magnitude of relative change, because these models reproduce overall patterns and natural trends, the performance of CMIP6 models concerning BA must be critically reflected (Hantson et al., 2020; Spafford and MacDougall, 2021). Here, we included all models providing *BurntFractionAll* and ESM tree fraction *treeFrac*. Furthermore, we assume

grasslands and other natural land as given by the IAM land cover / land use datasets to burn before forest is affected (threshold burning in Fig. S2).

Note that *BurntFractionAll* from the CMIP6 land models relies on different spatial patterns and grid cell shares of forest cover the IAM's projection. Hence, a transfer of this value and its application to IAM land cover and land use can only serve as first-order estimate and cannot replace a work-intense detailed vegetation modeling exercise with the IAM's land




cover projection as input, which is left for future work and out of scope for this perspective. While such choices of method (proportional vs. threshold burning, which other land covers to burn before forest, projecting *BurntFractionAll* directly on IAM land cover) seem to affect our results quantitatively, the overall finding of significantly up-sloping global areas of burned forest from A/R leading to increased risk ($10\% < \Delta_{rel}A_{pot.burned} < 100\%$ between 2020 and 2090) is robustly maintained under our sensitivity assessments.

## Appendix C: Forest area in land cover and land use data sets in past and present

This section gives an overview of the configurations and specifications of the models considered in this study (see Table C1), which can help to better understand the land use and land cover projections (see also suppl. Fig. S6).

In most IAMs the amount of A/R results from land use decisions based on cost minimization (Humpenöder et al., 2015;Hasegawa et al., 2017;Doelman et al., 2020). The presented ensemble of large-scale afforestation scenarios has shared assumptions en-
coded in the Shared Socioeconomic Pathway 1 (SSP1, (van Vuuren et al., 2017)) with the climate mitigation policy options SSP1-2.6 and SSP1-1.9. Most importantly these assumptions include a global price on carbon emissions and emission budgets in the mitigation scenarios that are roughly compatible with warming levels 2.0 and 1.5°C. Still, there are many differences in model setup and in the underlying assumptions remaining. The value of a growing forest carbon stock is implemented using a carbon price, which, depending on the model, is either determined in the energy sector model (Popp et al., 2017) or from a
comprehensive simulation of energy sector and land use (Krey et al., 2020; Wise et al., 2009). This price is designed to keep carbon emissions compatible with a certain carbon budget and a climate target (here with 1.5 or 2°C of global warming) across economic sectors. Changing land use from pasture to managed forest for example is rewarded with a price corresponding to the additional amount of carbon stored or expected to be stored. The equivalent is included in the computation as cost in case of deforestation and the corresponding drop in carbon density. Consequently, the level of the carbon price drives modeled A/R,
especially where potential revenues from forestry aside this price are comparable with or lower than those from other land uses. Ultimately, global estimates of A/R volume are sensitive to its local effectiveness to store carbon and to deliver multiple forest products (Humpenöder et al., 2014; Doelman et al., 2020; Frank et al., 2021).

In the model chain from IAMs to ESMs (Fig. 2), LUH was developed to harmonize historical land information and the land use outputs of IAMs for the use within ESMs (Hurtt et al., 2011). The second generation of LUH under CMIP6, LUH2,
provides land use and land cover change (LULCC) information in one classification for all scenarios, which had been computed by different models with different land use and land cover classifications (Hurtt et al., 2020). A harmonized data set fit to the original land use data sets was produced using the Global Land Model. While globally aggregated LULCC (e.g. global pasture or forest area change) were conserved, the spatial patterns of forest were not. This leads to LUH2 projections, also assessed here, showing different forest exposures to climate impacts such as fire weather than the IAM projections.
IMAGE and REMIND-MAgPIE include information on climate impacts from the vegetation model LPJmL (Doelman et al., 2020; Humpenöder et al., 2022). LPJmL computes in daily time steps and hence accounts for day-to-day variability. This allows the modeling of vegetation response to climate on sub-annual time scale via altered productivity and background mortality.



Natural disturbances, such as fire and heat stress for boreal trees are nevertheless modeled in annual time steps (Schaphoff et al., 2018; Braakhekke et al., 2019). In the IMAGE framework, LPJmL in an online mode informs the land use module annualy e.g. about carbon density using daily values emulated from annual climate provided by MAGICC, a simple climate model (Doelman et al., 2020). Fire impacts on carbon density are included in natural forest dynamics, but are excluded in forest plantations used for A/R. REMIND-MAgPIE is informed by crop and vegetation data modeled offline by LPJmL (Humpenöder et al., 2015), which is operated in daily time steps with daily climate data (Schaphoff et al., 2018). The limitation for the representation of vegetation impacts of climate extremes on sub-annual time scale therefore lies in the climate variables used and the physical processes represented in LPJmL. REMIND-MAgPIE offers an option to model fire emissions and heat stress of boreal forests but currently no additional explicit disturbance processes which lead to carbon losses. Typically, extreme conditions significantly lower the modeled productivity and enhance background mortality (Schaphoff et al., 2018). With a corresponding calibration (Forkel et al., 2019) LPJmL allows for an implicit representation of disturbances within the regimes provided by the calibration data sets. REMIND-MAgPIE uses annual potential carbon density, the maximum attainable carbon density of forests, which is modulated by changing climate in the model LPJmL and determined by aggregating from daily to annual values.

In GCAM-DEMETER and AIM the potential carbon stocks are represented much simpler, are not even spatially explicit, they are assumed to be constant within forest classes / economic regions.

The assessed versions of MESSAGE-GLOBIOM and GCAM-DEMETER only include climate impacts on croplands, not on forests. In MESSAGE-GLOBIOM this leads to significant shifts from forest to cropland in the comparison of a climate impact (in this model mainly $CO_2$-fertilization and mean climate, see Table 1) vs. a no climate impact scenario. This shows that differential treatment of land types can lead to imbalances in land use changes, in this case leading to less forestation on former cropland under a simulation including climate impacts on agriculture. To this effect, we refer to as "indirect impacts on forest allocation" in Table 1. Note that even this indirect tendency of less forestation under climate change, likely induced by productivity-enhancing climate impacts on cropland, is significantly stronger than variations generated by the six different climate models forcing the climate impacts (see suppl. Fig. S7, S12 and suppl. section S8).

Overall, IAMs and LUH2 have different initial land cover and land use inventories, starting dates, time steps, modeling procedures, spatial resolutions and classification schemes leading to differences both in past and present land use (Bayer et al., 2021; Brown et al., 2021). For example, as the only IAM in this ensemble, REMIND-MAgPIE includes national determined contributions (NDCs) in land use in the model. For the present study the gridded forest cover data was aggregated to the finest common resolution in space and time, 0.5°x0.5° and 10 years. The available forest classes were aggregated into one. The expansion of forest area in these afforestation scenarios is dominated by managed forestation in the data sets which provide such separate class.

Some forest management options beyond A/R might be at risk in a more fire-prone future climate. Four out of six IAM data sets account for rotations and forest age, but none of them has any mechanism of climate impacts on management in forestry.

We compare present-day values of forest area in IAM projections with the forest area in observational data sets to assess the plausibility of past and present land cover and land use. It should be noted though that there is also substantial disagreement



between observational data sets themselves. For example, satellite imagery can not distinguish all kinds of land cover and different classification schemes give different global tree cover areas (Harper et al., 2023; Buchhorn, 2020, Fig. S6). National

inventories combined in FAO forest resource assessments (FAO, 2020) give slightly higher values. Locally, i.e. at grid-cell scale ($\approx 0.5°$), the representation of forest cover in IAM data sets compared to satellite products is moderate to weak, owing to the substantial differences in ecosystem and landscape classification. Overall, the range of global forest areas in the IAM data sets for present day is compatible with observational data sets, while showing a much larger spread. In 2020, the first common date of the model outputs, the standard deviation of global forest area among the data sets amounts 3.2 Mkm$^2$. Under SSP1-2.6

it grows to 6.2 Mkm$^2$ over the course of the century.



**Table C1.** Configurations of land use models used in IAMs for the data sets included in this study.

| Dataset names according to fully coupled IAMs | LUH2 | AIM | GCAM-DEMETER | IMAGE | MESSAGE-GLOBIOM | REMIND-MAgPIE |
|---|---|---|---|---|---|---|
| Version and name of IAM, land-use component and vegetation input data sets | GLM2 (building on data from IMAGE 3.0). Gridded potential aboveground biomass is determined by Miami-LU model; land is potential forest when $\rho > 2$ kgC/m². | AIM-SSP/RCP Ver2018 Potential carbon density is input as constant per Agro-Ecological Zone. | GCAM v4.3.chen DEMETER v1.chen Potential carbon density is input as constant per plant functional type. | IMAGE 3.2 including fully coupled LPJmL 4 | GLOBIOM-G4M stand-alone (building on price and bioenergy demand from MESSAGE-GLOBIOM. Carbon density: G4M; Crop and pasture yields: EPIC | MAgPIE 4.4 (building on data from REMIND 2.1 fully coupled with MAgPIE 4.2; Potential carbon maps from LPJmL 4 for natural vegetation and LPJmL 5.2 for crops and pasture, National determined contributions in land use are included.) |
| Existing publication | Hurtt et al. (2020) | Fujimori et al. (2017) | Chen et al. (2022) Chen et al. (2019) | Doelman et al. (2022) Schaphoff et al. (2018) | Frank et al. (2021) Rogelj et al. (2018) | Humpenöder et al. (2022) |
| Land use model time step [a] | 1 | 10 | 5 | 1 | 10 | 5 until 2060, 10 from 2060 onward |
| Output time step [a] | 1 | 10 | 5 | 5 | 10 | 5 until 2060, 10 from 2060 onward |
| Initialization [year] | 2015 (for future land use) | 1960 | 1700 (GCAM), 1992 (DEMETER) | 1970 | 2000 | 1985 |
| Calibration until [year] | - | 2005 | 2005 | 2015 | 2015 | 2015 |
| Simulation end [year] | 2100 | 2100 | 2100 | 2100 | 2100 | 2100 |
| Land use model spatial resolution | 2.0°x2.0° | 17 regions | 384 (region-basin) units | 5'x5' | 37 regions, 2°x2° supply units | 12 regions, 2000 units |
| Downscaled output spatial resolution | 0.25°x0.25° | 0.25°x0.25° | 0.05°x0.05° | 5'x5' | 0.25°x0.25° | 0.5°x0.5° |
| Number of Land use / Land cover classes | 12+1* | 7 | 33 | 20 | 11 | 7 |
| Number of forest classes | 3 | 2 | 8 | 5 | 4 | 3 |
| Special class for managed forestation | Yes | Yes | No | Yes | Yes | Yes |
| Accounts for rotation and forest age | No | Yes | No | Yes | Yes | Yes |
| Includes climate impacts on rotation and forest age | No | No | No | No | No | No |

( * LUH2 treats added tree cover separated from other forest classes to fit forest expansion in the IMAGE SSP1-2.6 and SSP1-1.9 land use simulation.)



## Appendix D: Limitations

Having mentioned the limitations of our analysis underlying the perspective along the main text, we summarize them here for a concise overview and argue why our perspective holds under these limitations.

Our focus on land use projections by models, that provided land inputs to CMIP6, limits our analysis to these few most

prominent models. Our analysis is limited by the choice of climatic variables. Fire weather is a measure for fire hazard together with exposure, not fully capturing fire risk (in terms of e.g. carbon emissions), neither covering the entire spectrum of environmental stress. Using burned area from fire models within ESMs spans the way to fire impact but is heavily reliant on the assumptions around impact transfer from ESM to IAM land cover. More importantly, the performance of fire impacts from ESMs is very limited (Hantson et al., 2020; Spafford and MacDougall, 2021). However, as fire models in state-of-the-art

ESMs tend to underestimate fire impacts, our assessment remains a conservative one, hence this limitation does not affect the stringency of our main line of thought, namely the argument for more precise input and use of climate-related forest impacts in IAMs.

Additionally, we take feedbacks of A/R on climate and changing vulnerabilities into account only to a limited amount. For example, temperature and precipitation are expected to change in tropical regions with changed forest cover (Li et al.,

2022). While we do use the climate projections under SSP1-2.6, which include land cover change (and A/R) according to the LUH2 dataset, these do not match the A/R patterns of the IAMs exactly. However, as our analysis of the LUH2 forest cover does not show significantly lower danger from land cover change induced climate changes, we expect this to be a minor issue. Importantly, the identified signal of increasing danger, mainly driven by exposure increase, is extraordinarily strong and emphasizes the need to consider more climate information in land use projections. This is particularly relevant given the range

of other regional increases in climate extremes, such as heatwaves, droughts and heavy precipitation events, that are projected even at 1.5°C or 2°C of global warming (Seneviratne et al., 2021).



*Author contributions.* This research is part of the PhD work of FJ under supervision of SIS and JS at ETH Zurich. FJ, JS, YQ and SIS conceptualized the pespective and analysis. FJ performed the data analysis, produced the figures and drafted this manuscript. MW, FH, JD, SF, MWö and MG helped with land use and land cover data provision of the different Integrated Assessment Models. YQ calculated the FWI
from CMIP6 climate data. JS, YQ, SIS, MW, JD, SF, PH, FH, ALDA, CM, KBN, RSP, AP and DvV contributed to the interpretation of the analysis and the overall perspective and improved the manuscript.

*Competing interests.* At least one of the (co-)authors is a member of the editorial board of *Earth System Dynamics*. The authors also have no other competing interests to declare.

*Acknowledgements.* The authors acknowledge funding and support from the European Research Council Proof-of-Concept MESMER-X
project, under grant agreement No. 964013, and from the European Union's Horizon 2020 research and innovation program (PROVIDE project, grant agreement No. 101003687; RESCUE project, grant agreement No. 101056939; ForestNavigator project, grant agreement No. 101056875; including funding from the Swiss Secretariat for Education, Research and Innovation (SERI)). We acknowledge the Potsdam Institute for Climate Impact Research, PBL Netherlands Environmental Assessment Agency, the International Institute for Applied Systems Analysis and the Joint Global Change Research Institute for the production and provision of the land use and land cover data. Data on
burned area and tree fractional coverage produced under CMIP6 are provided by WCRP (World Climate Research Program) and ESFG (Earth System Grid Federation). To improve the linguistic expression in some passages within this manuscript, ChatGPT was used. We acknowledge the development and provision of the python packages and wrappers maplotlib (Hunter, 2007), numpy (Harris et al., 2020), proplot (Davis, 2021), scipy (Virtanen et al., 2020) and xarray (Hoyer and Hamman, 2017). We thank Louise Parsons Chini, Shinichiro Fujimori, George Hurtt, Andreia Ribeiro and Marshall Wise as well as the three anonymous reviewers for very helpful comments.



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
