# Peer review of "Fire Weather Compromises Forestation-reliant Climate Mitigation Pathways"

_EGUsphere, 2024_

## Author Comment (AC1)

Referee comments to *Fire Weather Compromises Forestation-reliant Climate Mitigation Pathways*

Referee #1:
https://egusphere.copernicus.org/preprints/egusphere-2024-15#RC1

1.0 Carbon sink is a crucial strategy for mitigating the effects of climate change, but wildfires have multiple perturbations to vegetation carbon storage. The authors simulated forestation-dependent climate mitigation scenarios using five integrated assessment models. The burned area caused by fire weather was projected over future forested regions. While the topic is very interesting and relevant to the scope of EGUsphere, the MS suffers from several major weaknesses that need to be addressed.

Thank you for this summary. As you mention later, you consider as a major weakness that such discrepancies hinder the credibility of these transformation pathways. However, we want to clarify that this is precisely our point: we use existing IAMs data and CMIP6 Earth System Model projections to confront them, and we exhibit lacks of consistency affecting the plausibility of the forestation scenarios. We thank you for acknowledging the relevance of our task.

Introduction:

1.1.1 The literature review section should not simply list what the other researchers have done. I suggest that the authors briefly discuss how the problems addressed by previous studies advance the research topic here.

Thank you for this observation. Upon your suggestion we will enhance the connection between the references to provide a comprehensive introduction to the debate on the feasibility of large-scale forestation under future climate conditions.

1.1.2 The authors thoroughly discussed the mechanisms of wildfires affecting carbon sinks. It would be useful to add more recent data on how much of the global carbon sink is affected by wildfires.

Thanks for this suggestion, this would be a good way to introduce the reader to a quantitative perspective on carbon dynamics under fire. As the Global Carbon Budget 2023 (Friedlingstein et al., 2023) details, a direct comparison of fire emissions with data on the land carbon sink at the global scale is not adequate as currently measured fluxes are a mixture of gross and net fluxes, hiding the role of natural disturbance-recovery cycles vs. net losses beyond such natural cycles (section 3.8.3). However, we will use recent ressources (e.g. Fan et al., 2023) to give this important perspective as far as possible with state-of-the-art data.

Friedlingstein, P., et al.: Global Carbon Budget 2023, Earth System Science Data, 15, 5301–5369, https://doi.org/10.5194/essd-15-5301-2023, 2023

Fan, L., Wigneron, J.-P., Ciais, P., Chave, J., Brandt, M., Sitch, S., Yue, C., Bastos, A., Li, X., Qin, Y., Yuan, W., Schepaschenko, D., Mukhortova, L., Li, X., Liu, X., Wang, M., Frappart, F., Xiao, X., Chen, J., Ma, M., Wen, J., Chen, X., Yang, H., van Wees, D., and Fensholt, R.: Siberian carbon sink reduced by forest disturbances, Nature Geoscience, 16, 56–62, https://doi.org/10.1038/s41561-022-01087-x, 2023.

1.1.3 The last paragraph: 'How to set up and how to improve the experiment' is not an appropriate scientific question. I suggest reorganizing this paragraph.

Thank you for this comment. We would like to reemphasize that with this perspective we try to shed light on how modeling procedures are currently set up. We think that discussing process representation and how it is currently limiting reliability, interpretability and scientific progress is

scientifically essential questioning. To clarify our ambitions and questions, we will reformulate the paragraph.

Results:

1.2.1 The 2090 fire risk for MESSAGE-GLOBIOM and REMIND-MAgPIE is lower than that during 2050 in Figure 4a, but in Figure 4b the results are reversed. It is confusing.

Thank you for pointing this out. Displaying relative changes of all forests in panel b vs. absolute values in panel a for A/R regions only (upper boxes) can lead to confusing perception. These upper boxes indicate the mean FWI in forestation regions.

As an example we here point out the case of MESSAGE-GLOBIOM. In this IAM, all forests show a mean danger of approx. 16 index points in 2020 (broad lower box in a).

Once forestation in very highly hazardous regions until 2050 (regional mean danger 22 index points, left upper box in panel a) is in place, the global mean forest danger increases by 5% (panel b).

As the forestation area until 2090 is added in highly hazardous (regional mean danger 20 index points, right upper box in panel a), the global mean forest danger increases further but only by another 1% (right half of panel b).

While we think that this way is well representing the data, we will make these details much clearer by adapting the caption text and the explanations in the main text.

1.2.2 The multi-model results reveal a less than 35% increase of FWI in 2090, but the increase in the burned area could be close to 80% (Figure 4a, b).

Your observation that the relative increase in burned areas (BA) is significantly higher and more uncertain than the relative increase in area-weighted FWI (danger) is an important detail. In our understanding it arises from two features: first, the BA in response to warming and drying trends naturally is more uncertain than the exposure. In our estimation procedure, this is due to spatially more heterogeneous coverage of the BA as ESM output than FWI. Naturally, this arises from heterogeneous vulnerability and ignition sources (see e.g. Jones et al., 2022). Second, fire response to drivers like dryness in some regions have been found to follow non-linear relationships (e.g. Zheng et al., 2023, Fig.4c on fire emission response).

1.2.3 If planted trees are greatly likely to be at risk of fire, are they still needed? If they are needed, how can they be protected from fire based on your results? I suggest the authors briefly discuss it.

Questioning the necessity of A/R under these circumstances might seem reasonable. However, a complete answer to this question cannot be given in this work, because it goes far beyond the limits of this manuscript. As a short answer, re/a-forestation may be necessary to mitigate climate change, but it has to be done carefully.

This means choosing the right locations, adequate species, ecologically adequate fire management, etc. The purpose of this manuscript is to show that this should not be ommited in IAMs in the pursuit of more robust transformation pathways. The decision on whether or not A/R still is useful under projected FWI and burned area increases depends on the comparative evaluation of other services, hazards and vulnerabilities of A/R. In the framework of IAMs, when carbon loss from disturbances like fire and costs of fire prevention is attempted to be internalized, realized A/R

allocation might decrease. In assessments including aspects beyond quantifiable measures however, the picture might look more severe. Not at every place where IAMs foresee forestation, forest establishment, and in particular senseful fire prevention is plausible. We would like to refer you to lines 116 - 118 (risk reduction by exclusion of highly exposed areas), 120 - 124 (fire prevention potential), 178 - 180 (internalization of services and fire prevention in IAMs), where we briefly discuss such aspects.

Jones, M. W., Abatzoglou, J. T., Veraverbeke, S., Andela, N., Lasslop, G., Forkel, M., Smith, A. J. P., Burton, C., Betts, R. A., van der Werf, G. R., Sitch, S., Canadell, J. G., Santín, C., Kolden, C., Doerr, S. H., and Le Quéré, C.: Global and Regional Trends and Drivers of Fire Under Climate Change, Reviews of Geophysics, 60, e2020RG000 726, https://doi.org/10.1029/2020RG000726, 2022.

Zheng, B., Ciais, P., Chevallier, F., Yang, H., Canadell, J. G., Chen, Y., van der Velde, I. R., Aben, I., Chuvieco, E., Davis, S. J., Deeter, M., Hong, C., Kong, Y., Li, H., Li, H., Lin, X., He, K., and Zhang, Q.: Record-high $CO_2$ emissions from boreal fires in 2021, Science, 379, 912–917, https://doi.org/10.1126/science.ade0805, 2023.

1.2.4 There are large differences between the projected and observed forest areas in 2020 (Figure 1). There are also large differences in the magnitudes and patterns of projections from different models (Figure 5), although they all show an upward risk. How can these results be credible?

All these IAM simulations had been performed prior to and independent of this study. Assessing them, we find their A/R pathways showing substantial differences both in planted forest distribution and volume and resulting changes in average exposure and danger. This is precisely what we want to show, that the transformation pathways in the literature lack in credibility on this aspect: the plausibility of forestation according to IAMs' SSP1-2.6 is low. Yet it remains that this way of modeling land use is state of the art while in the process of being improved. To foster this improvement, we discuss the spread and diversity we found in these existing simulations. It is owed to diverse assumptions, initializations and modeling methodologies (see Appendix C for a detailed discussion).

1.2.5 Figure 5 sets values from -0.5 to 0.5 to the same color, which does not help distinguish the areas of decreasing FWI.

Thank you for this detailed observation. We will alter the color bar such that we keep transparency high close to zero change.

---

## Author Response (AR1)

**Author's Response**

to Referee comments to *Fire Weather Compromises Forestation-reliant Climate Mitigation Pathways*

Referee #1:
https://egusphere.copernicus.org/preprints/egusphere-2024-15#RC1

1.0 Carbon sink is a crucial strategy for mitigating the effects of climate change, but wildfires have multiple perturbations to vegetation carbon storage. The authors simulated forestation-dependent climate mitigation scenarios using five integrated assessment models. The burned area caused by fire weather was projected over future forested regions. While the topic is very interesting and relevant to the scope of EGUsphere, the MS suffers from several major weaknesses that need to be addressed.

Thank you for this summary. We analyze whether the effectiveness of forestation in the current spatially explicit representation of IAM projections could be compromised by fire disturbance, based on available independently generated scenarios from a range of state-of-the-art IAMs. We thank you for acknowledging the relevance of our task.

Introduction:

1.1.1 The literature review section should not simply list what the other researchers have done. I suggest that the authors briefly discuss how the problems addressed by previous studies advance the research topic here.

We appreciate the comment and modified the introduction accordingly. Previous studies highlight the present and projected relevance of fire for the land C sink, but its under representation in IAMs has not been discussed until now.

> Lines 38-44: "A debate about the conditions and feasibility of large scale A/R under projected future climate conditions is therefore needed. Comparing maps of fire weather change and forestation potential Hermoso et al., 2021 recently have already brought forward their concern on increasing fire disturbance of forest restoration projects in Europe. Meanwhile, the science of drivers of future fire risk for forest carbon sinks, e.g. Clarke et al. (2022) identifying high water vapor pressure deficit as major threat, goes on. Arguing against major concerns, Golub et al. (2022) presented a land use allocation assessment that calculates die-back rates for various biomes as a function of changes in global mean temperature. Their findings suggest that wildfires may not compromise forest-based climate strategies"

1.1.2 The authors thoroughly discussed the mechanisms of wildfires affecting carbon sinks. It would be useful to add more recent data on how much of the global carbon sink is affected by wildfires.

Thanks for this suggestion. We now introduce the reader to a quantitative perspective on carbon dynamics under fire with recent estimates from the Global Carbon Budget 2023 (Friedlingstein et al., 2023). It is important to note that a direct comparison of fire emissions with the land carbon sink at the global scale is not adequate as currently measured fluxes are a mixture of gross and net fluxes, hiding the role of natural disturbance-recovery cycles vs. net losses beyond such natural cycles (section 3.8.3 in Friedlingstein et al., 2023).

Lines 25-30: "Fire as a prominent hazard for carbon accumulation in forests is influenced by the increasing intensity and frequency of climate extremes (Seneviratne et al., 2021). Gross fire emissions amounted to 1.8 Gt C yr$^{-1}$ on average between 2003 and 2022, and reached 1.9-2.3 Gt C yr$^{-1}$ in 2023 (Friedlingstein et al., 2023). Although only part of these emissions are from climate-change driven fires, and a post-fire vegetation recovery sink is unaccounted for, it is important to note the relevance of fire for the land carbon sink. For example, anomalously high fire emissions, as those from recent boreal forest fires, could contribute to large biomes ceasing to act as carbon sinks (Fan et al., 2023)"

Friedlingstein, P., et al.: Global Carbon Budget 2023, Earth System Science Data, 15, 5301–5369, https://doi.org/10.5194/essd-15-5301-2023, 2023

Fan, L., Wigneron, J.-P., Ciais, P., Chave, J., Brandt, M., Sitch, S., Yue, C., Bastos, A., Li, X., Qin, Y., Yuan, W., Schepaschenko, D., Mukhortova, L., Li, X., Liu, X., Wang, M., Frappart, F., Xiao, X., Chen, J., Ma, M., Wen, J., Chen, X., Yang, H., van Wees, D., and Fensholt, R.: Siberian carbon sink reduced by forest disturbances, Nature Geoscience, 16, 56–62, https://doi.org/10.1038/s41561-022-01087-x, 2023.

1.1.3 The last paragraph: 'How to set up and how to improve the experiment' is not an appropriate scientific question. I suggest reorganizing this paragraph.

Thank you for this comment. We would like to reemphasize that with this perspective we try to shed light on how modelling procedures are currently set up. To clarify our ambitions and questions, we reformulated the paragraph to give an overview not only on what we are asking about but also on how we are answering these important questions:

Lines 50-61: "In this perspective, we therefore assess the plausibility of forestation-reliant climate mitigation scenarios as used within CMIP6. We do this by analyzing whether the effectiveness of forestation in the current spatially explicit representation of IAM projections could be compromised by fire disturbance, based on available scenarios from a range of state-of-the-art IAMs. First, we present an overview of the amounts of forestation as projected by IAMs and how sensitive they are to climate change effects (Section 2). In Section 3, we introduce a climatological measure for atmospheric pressure on forests to burn, seasonally extreme fire weather index, and analyze its behavior in Earth System Models (ESMs) during both historical and future periods for comparison with climate data. By combining these datasets, we evaluate the extent and locations of increased exposure of global forests to fire weather, discerning the relative contributions of exposure change (forest expansion) and hazard change (fire intensification due to climate change) to the rise in forest fire danger. (Section 4). To put this into perspective, we provide an overview of the modeling landscape on how spatially explicit information about forest disturbances and climate change is treated in state-of-the-art land use allocation in IAMs (Section 5). Finally, we discuss how the representation of climate impacts on forestation can be improved to arrive at more substantiated climate mitigation scenarios (Section 6)."

Results:

1.2.1 The 2090 fire risk for MESSAGE-GLOBIOM and REMIND-MAgpIE is lower than that during 2050 in Figure 4a, but in Figure 4b the results are reversed. It is confusing.

Thank you for pointing this out. Displaying relative changes of all forests in panel b vs. absolute values in panel a for A/R regions only (upper boxes) can lead to confusing perception. These upper boxes indicate the mean FWI in forestation regions.

As an example we here point out the case of MESSAGE-GLOBIOM. In this IAM, all forests show a mean danger of approx. 16 index points in 2020 (broad lower box in a).

Once forestation in very highly hazardous regions until 2050 (regional mean danger 22 index points, left upper box in panel a) is in place, the global mean forest danger increases by 5% (panel b).

As the forestation area until 2090 is added in highly hazardous (regional mean danger 20 index points, right upper box in panel a), the global mean forest danger increases further but only by another 1% (right half of panel b).

While we think that this way is well representing the data, we mde these details much clearer by adapting the explanations in the main text.

> Lines 108-115: "Over the course of the century, the projected forested regions in IAMs are increasingly under danger from fire weather (relative increase in $FWI_F$ of up to more than 30 %, Fig. 4b).
> This finding, which is robust against inter-ESM differences in fire weather, has two reasons: First, fire weather intensifies through heating and drying trends over forest areas that already exist in 2020 as a consequence of climate change (hazard as driver). Second, and more important, most of the increase in danger under SSP1-2.6 is driven by A/R in regions of already high and/or intensifying fire weather (exposure as driver, see narrow boxes in Fig. 4a higher than broad boxes and darker higher than lighter bars in Fig. 4c, Fig. S5 for spatially explicit contributions, S10 for the scenario SSP1-19 and Section S2 for explanatory text and formula)."

1.2.2 The multi-model results reveal a less than 35% increase of FWI in 2090, but the increase in the burned area could be close to 80% (Figure 4a, b).

Your observation that the relative increase in burned areas (BA) is significantly higher and more uncertain than the relative increase in area-weighted FWI (danger) is an important detail. In our understanding it arises from two features: first, the BA in response to warming and drying trends naturally is more uncertain than the exposure. In our estimation procedure, this is due to spatially more heterogeneous coverage of the BA as ESM output than FWI. Naturally, this arises from heterogeneous vulnerability and ignition sources (see e.g. Jones et al., 2022). Second, fire response to drivers like dryness in some regions have been found to follow non-linear relationships (e.g. Zheng et al., 2023, Fig.4c on fire emission response). We have already detailed on this in Appendix B and refer the reader to this and the related supplement in line 123 in the main text.

1.2.3 If planted trees are greatly likely to be at risk of fire, are they still needed? If they are needed, how can they be protected from fire based on your results? I suggest the authors briefly discuss it.

Questioning the necessity of A/R under these circumstances might seem reasonable. However, a complete answer to this question cannot be given in this work because it goes far beyond the limits of this manuscript. As a short answer, re/a-forestation may be necessary to mitigate climate change, but it has to be done carefully, and its potential should indeed not be overestimated.

This means choosing the right locations, adequate species, ecologically adequate fire management, etc. The purpose of this manuscript is to show that this should not be omitted in IAMs in the pursuit of more robust transformation pathways. The decision on whether A/R is still useful under projected FWI and burned area increases depends on the comparative evaluation of other services, hazards and vulnerabilities of A/R. In the framework of IAMs, when carbon loss from disturbances like fire and costs of fire prevention is attempted to be internalized, realized A/R allocation might decrease. In assessments including aspects beyond quantifiable measures however, the picture might look more severe. Not at every place where IAMs foresee forestation, forest establishment, and in particular senseful fire prevention is plausible. We would like to refer you to lines 129 – 131 (formerly lines 116 – 118, risk reduction by exclusion of highly exposed areas),

> "If areas for forestation were excluded when located in regions of extreme fire regimes (FWI > 47 in 2090, 95th percentile of historical conditions), the overall forestation would be reduced by 4 % (IMAGE) to 20 % (AIM) in 2090, clearly illustrating the potential impact of fire danger on forestation allocation."

133 – 137 (formerly 120 – 124, fire prevention potential),

> "Although forests exposed to high FWI are not guaranteed to burn more intensely or more often in every location, long-term and large-scale average response in this direction is very likely, particularly under the absence of fire prevention measures (IAMs do not include the costs for these to date). Such fire prevention may be effective on the medium-term range (Wu et al., 2021), but often unsustainable on longer time scales (Moya et al., 2019; Bowman et al., 2009)."

and lines 191-193 (formerly 178 – 180, internalization of services and fire prevention in IAMs),

> "Additionally internalizing the expected increase in costs in fire prevention measures and changes to ecosystem services from e.g. biodiversity could help to make forestation projections more comprehensive (Pawson et al., 2013; Hansen et al., 2001; Prestele et al., 2016)."

where we already in the original manuscript briefly discuss such aspects.

Jones, M. W., Abatzoglou, J. T., Veraverbeke, S., Andela, N., Lasslop, G., Forkel, M., Smith, A. J. P., Burton, C., Betts, R. A., van der Werf, G. R., Sitch, S., Canadell, J. G., Santín, C., Kolden, C., Doerr, S. H., and Le Quéré, C.: Global and Regional Trends and Drivers of Fire Under Climate Change, Reviews of Geophysics, 60, e2020RG000 726, https://doi.org/10.1029/2020RG000726, 2022.

Zheng, B., Ciais, P., Chevallier, F., Yang, H., Canadell, J. G., Chen, Y., van der Velde, I. R., Aben, I., Chuvieco, E., Davis, S. J., Deeter, M., Hong, C., Kong, Y., Li, H., Li, H., Lin, X., He, K., and Zhang, Q.: Record-high CO 2 emissions from boreal fires in 2021, Science, 379,912–917, https://doi.org/10.1126/science.ade0805, 2023.

1.2.4 There are large differences between the projected and observed forest areas in 2020 (Figure 1). There are also large differences in the magnitudes and patterns of projections from different models (Figure 5), although they all show an upward risk. How can these results be credible?

All these IAM simulations had been performed prior to and independent of this study. Assessing them, we find their A/R pathways showing substantial differences both in planted forest distribution and volume and resulting changes in average exposure and danger. This is precisely what we want to show, that the transformation pathways in the literature lack in credibility on this aspect: the plausibility of forestation according to IAMs' SSP1-2.6 is low. Yet it remains that this way of modeling land use is state of the art while in the process of being improved. To foster this improvement, we discuss the spread and diversity we found in these existing simulations. It is owed to diverse assumptions, initializations and modeling methodologies (see Appendix C for a detailed discussion).

1.2.5 Figure 5 sets values from -0.5 to 0.5 to the same color, which does not help distinguish the areas of decreasing FWI.

Thank you for this detailed observation. We altered the color bar such that we keep transparency high close to zero change.

Referee #2:
https://egusphere.copernicus.org/preprints/egusphere-2024-15#RC2
General:

2.0 This paper investigates the feedback mechanisms between climate change and forest carbon sequestration across five integrated assessment models (IAMs), where wildfires play a significant role in influencing the accumulation of carbon within forests. The Canadian Fire Weather Index (FWI) is a pivotal metric to assess potential fire risk. The subject matter is particularly intriguing for discussion, given that contemporary research tends to concentrate more on either afforestation efforts or the impacts of exacerbated wildfire weather conditions. Several minor points need to be addressed to elucidate the central concept more effectively.

Thank you for acknowledging both the relevance of our perspective as well as its interdisciplinary approach.

2.1 The introductory section outlines the advancements and constraints of current research, but seems inadequacy to explicitly establish strong relationships among these various aspects.

Thank you for this feedback, which agrees with the comment of referee #1 (1.1.1). In response, we enhanced the links and connections to guide the reader into current debate on the topic (see response to 1.1.1).

2.2 In Section 3, the authors employ FWI instead of alternative fire impact metrics due to its dependency solely on atmospheric conditions, rendering it more resilient. Moreover, FWI exhibits a positive interannual correlation across various forested regions. However, it is confusing that the calculated weighted mean value does not provide adequate support for the role FWI plays in serving as a proxy for wildfire potential. To improve this, it would be beneficial if the authors:

- Conduct a comparative analysis between FWI and other existing fire weather index systems.

- Concentrate their focus on forest areas that display notably high correlations.

Thank you for this highly valuable comment. Local FWI cannot fully represent fire risk, because vulnerability and exposure of the forest would matter as well. However, the same issue remains with alternative fire weather indices. Nevertheless, the FWI, as well as other fire indices, are useful to assess the physical hazard of changing weather conditions. Supported by extensive literature on the robust links between fire weather and fire occurrence (e.g. Abatzoglou et al, 2018; Bedia et al, 2015; Jones et al., 2022), we are confident that assuming constant and homogeneous vulnerability globally allows for conclusions on the globally aggregated level for overall relative trends

Following your helpful advice we assessed whether evaluating relative changes in FWI in regions with high correlation yields different results (see reply Fig. R1 here, also provided as supplementary Fig. S13).

We tested whether restricting the FWI analysis given in Section 4 of the perspective (in particular the relative increases in fire danger given in Fig. 4b,c) to areas where ESMs show a strong correlation $R_{min}$ between FWI and BA.

We find that changes in mean FWI do typically not differ by more than 4 % (Figure R1/ S13). Only if selecting just those pixels with a very high correlation (Rmin > 0.8) changes can differ by up to 7 %. Since the increase in mean FWI is

[Figure]

Fig. R1: Difference of the values for relative change of mean FWI given in Figure 4b and c, when the change in forest-area weighted mean response is calculated from grid cells showing higher then threshold $R_{min}$ correlation of FWI and BA in the 4 ESMs shown in Fig. S1 rather than from all grid cells as in Fig. 4b. The black dashed line indicates no difference to the unconstrained analysis.

similar or even higher for highly correlated areas, this underlines our previous findings and conclusions and was included as a new insight in lines 110-111 of the updated manuscript.

Abatzoglou, J. T., Williams, A. P., Boschetti, L., Zubkova, M., and Kolden, C. A.: Global patterns of interannual climate–fire relationships, Global Change Biology, 24, 5164–5175, https://doi.org/10.1111/gcb.14405, 2018.

Bedia, J., et al.: Global patterns in the sensitivity of burned area to fire-weather: Implications for climate change, Agricultural and Forest Meteorology 214-215, 369-379, https://doi.org/10.1016/j.agrformet.2015.09.002, 2015.

Jones, M. W., Abatzoglou, J. T., Veraverbeke, S., Andela, N., Lasslop, G., Forkel, M., Smith, A. J. P., Burton, C., Betts, R. A., van der Werf, G. R., Sitch, S., Canadell, J. G., Santín, C., Kolden, C., Doerr, S. H., and Le Quéré, C.: Global and Regional Trends and Drivers of Fire Under Climate Change, Reviews of Geophysics, 60, e2020RG000 726, https://doi.org/10.1029/2020RG000726, 2022.

2.3 Several studies (ref1, ref2) have demonstrated that anticipated changes in fire weather and ongoing wildfire activities can drive long-term shifts in forest species composition and lead to significant transitions from woody vegetation cover to less dense vegetative types such as scrubland and grassland. In this context, I am interested to know how the IAMs in this paper address or account for these vegetation dynamics and their implications on the ecosystem.

In some IAMs, vegetation dynamics and PFT composition are only modelled by separate vegetation models like LPJmL. In others, simpler, static vegetation maps and parameters are used (see Appendix C for overview). Whereas REMIND-MAgPIE does include LPJmL data in an offline mode, IMAGE simulation runs include LPJmL vegetation dynamics coupled to other socio-economic processes annually. Advanced processes like aridity limiting post-fire recovery have not

been assessed with LPJmL yet. However, with its modeling structure LPJmL is generally capable to model landscapes and vegetation composition responding to disturbance (e.g. Ostberg et al., 2015). We added this information in Appendix C:

> Lines 299-301: "Although this has not been assessed for all circumstances of post-disturbance vegetation dynamics,LPJmL is principle has the mechanisms implemented to represent some climate-specific vegetation response to disturbance Ostberg et al. (2015)."

It is important to contextualize that the land model components of IAMs are land use models optimizing the allocation of land use and land cover change according to optimization of internalized profits and costs. Shifting vegetation suitability is therefore rather a boundary condition than a driver of the projected land cover changes (compare Table 1 and lines 158 – 165, formerly 145 – 152, see also Schaphoff et al., 2018 and Braakhekke et al., 2019).

> Lines 158-165: "Among the here analyzed models, only IMAGE (existing forests, not in A/R areas) and REMIND-MAgPIE (all forests) account for carbon losses from changing climate conditions including fire in their estimates of forest carbon density. IMAGE and REMIND-MAgPIE include information on climate impacts from the vegetation model LPJmL (Doelman et al., 2020; Humpenöder et al., 2022), which accounts for day-to-day variability but does not include natural disturbances like fire on this timescale (Schaphoff et al., 2018; Braakhekke et al., 2019). In other IAMs, such processes are not included. Therefore, these models are expected to produce overly optimistic estimates of A/R effectiveness. Except IMAGE 3.0, the IAM providing the SSP1 simulations (Stehfest et al., 2014; O'Neill et al., 2016), neither the model versions used for projections under CMIP6 nor LUH2 included any climate feedbacks."

Ostberg, S., Schaphoff, S., Lucht, W., and Gerten, D.: Three centuries of dual pressure from land use and climate change on the biosphere, Environmental Research Letters, 10, 044 011, https://doi.org/10.1088/1748-9326/10/4/044011, 2015.

Schaphoff, S., Bloh, W. V., Rammig, A., Thonicke, K., Biemans, H., Forkel, M., Gerten, D., Heinke, J., Jägermeyr, J., Knauer, J., Langer-wisch, F., Lucht, W., Müller, C., Rolinski, S., and Waha, K.: LPJmL4 - A dynamic global vegetation model with managed land - Part 1: Model description, Geoscientific Model Development, 11, 1343–1375, https://doi.org/10.5194/gmd-11-1343-2018, 2018.

Braakhekke, M. C., Doelman, J. C., Baas, P., Müller, C., Schaphoff, S., Stehfest, E., and Vuuren, D. P. V.: Modeling forest plantations for carbon uptake with the LPJmL dynamic global vegetation model, Earth System Dynamics, 10, 617–630, https://doi.org/10.5194/esd-10-617-2019, 2019.

2.4 In figure 5, the value between -0.5 and 0.5 was set to the same group in the color bar, demonstrating very little information.

Thank you, as requested also by referee #1 (1.2.4), we alter the color bar such that we keep transparency high close to zero change.

Ref1. Mekonnen, Z.A., Riley, W.J., Randerson, J.T. et al. Expansion of high-latitude deciduous forests driven by interactions between climate warming and fire. Nat. Plants 5, 952–958 (2019). https://doi.org/10.1038/s41477-019-0495-8

Ref2. Baudena, M., Santana, V.M., Baeza, M.J., Bautista, S., Eppinga, M.B., Hemerik, L., Garcia Mayor, A., Rodriguez, F., Valdecantos, A., Vallejo, V.R., Vasques, A. and Rietkerk, M. (2023), Increased aridity drives post-fire recovery of Mediterranean forests towards open shrublands. New Phytol, 239: 2416-2417. https://doi.org/10.1111/nph.19012